# Thyroid-Modulating Activities of Olive and Its Polyphenols: A Systematic Review

**DOI:** 10.3390/nu13020529

**Published:** 2021-02-06

**Authors:** Kok-Lun Pang, Johanna Nathania Lumintang, Kok-Yong Chin

**Affiliations:** 1Department of Pharmacology, Faculty of Medicine, Universiti Kebangsaan Malaysia, Jalan Yaacob Latif, Bandar Tun Razak, Cheras 56000, Kuala Lumpur, Malaysia; pangkoklun@ukm.edu.my; 2Faculty of Applied Sciences, UCSI University Kuala Lumpur Campus, Jalan Menara Gading, Taman Connaught, Cheras 56000, Kuala Lumpur, Malaysia; lumintang_johanna@yahoo.com

**Keywords:** olive oil, olive polyphenol, oleuropein, hydroxytyrosol, triiodothyronine, thyroxine

## Abstract

Olive oil, which is commonly used in the Mediterranean diet, is known for its health benefits related to the reduction of the risks of cancer, coronary heart disease, hypertension, and neurodegenerative disease. These unique properties are attributed to the phytochemicals with potent antioxidant activities in olive oil. Olive leaf also harbours similar bioactive compounds. Several studies have reported the effects of olive phenolics, olive oil, and leaf extract in the modulation of thyroid activities. A systematic review of the literature was conducted to identify relevant studies on the effects of olive derivatives on thyroid function. A comprehensive search was conducted in October 2020 using the PubMed, Scopus, and Web of Science databases. Cellular, animal, and human studies reporting the effects of olive derivatives, including olive phenolics, olive oil, and leaf extracts on thyroid function were considered. The literature search found 445 articles on this topic, but only nine articles were included based on the inclusion and exclusion criteria. All included articles were animal studies involving the administration of olive oil, olive leaf extract, or olive pomace residues orally. These olive derivatives were consistently demonstrated to have thyroid-stimulating activities in euthyroid or hypothyroid animals, but their mechanisms of action are unknown. Despite the positive results, validation of the beneficial health effects of olive derivatives in the human population is lacking. In conclusion, olive derivatives, especially olive oil and leaf extract, could stimulate thyroid function. Olive pomace residue is not suitable for pharmaceutical or health supplementation purposes. Therapeutic applications of olive oil and leaf extract, especially in individuals with hypothyroidism, require further validation through human studies.

## 1. Introduction

The olive tree, scientifically known as *Olea europaea* L., is mainly found in Mediterranean countries. It can grow and survive even in soils with low fertility during the drought and semi-drought seasons [1]. Olive fruit can be consumed as table olives after curing and fermentation to remove their bitterness [1,2]. On the other hand, olive oil is the main lipid source in the Mediterranean diet [3,4]. In epidemiological studies, the Mediterranean diet has been associated with reduced risk of developing cancer and cardiovascular and musculoskeletal diseases [5,6,7,8]. These health-beneficial effects are partially attributed to olive oil, which is used commonly in the Mediterranean diet [9,10,11,12].

According to the International Olive Oil Council and United States Department of Agriculture, olive oil can be graded according to the listed quality and purity criteria, which include colour, odour, flavour, free fatty acid content or acidity, and trans fat content [13,14]. Virgin olive oil (VOO) is generally extracted by cold extraction via mechanical processes like grinding, malaxation, recovery, and filtration [3,15]. Extra-virgin olive oil (EVOO) is the highest grade of VOO, having not more than 0.8% acidity, while olive oil with not more than 2% acidity is classified as VOO [13,14]. Refined olive oil is produced when the extracted olive oil undergoes physical or chemical (non-solvent)-based refining processes [16]. During these processes, most antioxidants and phenolics in the olive oil are removed [16,17]. The olive pomace or olive pulp solid is the olive paste leftover after the cold extraction process. Refined olive pomace oil is produced from the oil extracted with chemical solvents from the olive pomace [16,18]. Refined olive oil and olive pomace oil are eventually blended with VOO or EVOO and marketed as olive oil and olive pomace oil, respectively [18].

The major constituents of olive oil are triacylglycerols (98–99%), while the remaining contents are free fatty acids, glycerol, phosphatides, pigments, tocopherol, phenolic acid, sterol, and other microconstituents [11,19,20]. The major fatty acid in olive oil triacylglycerols is oleic acid (monounsaturated fatty acid), followed by linoleic acid, palmitic acid, and other minor fatty acids [11]. Despite the high level of triacylglycerols in olive oil, the beneficial effects of olive oil result from the microconstituents, including secoiridoid glycosides, phenolics, and flavonoids (apigenin, kaempferol, and luteolin) [20]. Oleuropein is the main secoiridoid glycoside in unripe olive fruit, and it contributes to the bitter taste of olive oil and table olives [21]. Hydroxytyrosol (3,4-dihydroxyphenylethanol, 3,4-DHPEA) and tyrosol (p-hydroxyphenylethanol, p-HPEA) are the most abundant phenolic alcohols in olive oil. They are converted from oleuropein through hydrolysis; thus, their concentrations increase with the ripening of olive fruit [1]. Oleuropein is also converted into oleuropein aglycone (3,4-dihydroxyphenylethanol-elenolic acid ester, 3,4-DHPEA-EA) via deglycosylation [22,23]. Oleocanthal (p-hydroxyphenylethanol-elenolic acid dialdehyde, p-HPEA-EDA) is the minor secoiridoid that gives the unique burning sensation of EVOO [4,24]. Other minor compounds, such as verbascoside, ligstroside aglycone (p-hydroxyphenylethanol-elenolic acid ester, p-HPEA-EA), and lignans, are also detected in EVOO [25,26].

Olive oil, especially EVOO, has been reported to possess several biological activities, such as antimicrobial, antiviral, antioxidant, anticancer, and osteoprotective actions, which have not been observed in other vegetable oils [7,27,28,29,30,31,32]. In addition, EVOO also contains the highest level of olive microconstituents like phenolics and secoiridoids [11], which are responsible for these biological activities. Olive leaf extracts also possess similar biological activities to olive oils, including anticancer, antidiabetic, antioxidant, and anti-inflammatory actions [33,34,35,36,37]. A phytochemical analysis revealed that olive leaf extracts contain similar phenolics and phytochemicals to VOO, mainly oleuropein, hydroxytyrosol, luteolin, and their derivatives [22,38,39,40]. Olive leaf phytochemicals like oleuropein are present in much higher concentrations than in olive oil [41,42,43]. Additionally, in vitro and in vivo experiments have shown that oleuropein, hydroxytyrosol, and tyrosol are potent antioxidant, anticancer, chemopreventive, and antiatherosclerotic agents [44,45,46,47,48,49,50,51,52,53].

Accumulated evidence suggests that bioactive phenolics in olive oil and olive leaf extracts possess hormone-modulating properties [54,55,56,57,58]. Olive oil and oleuropein were shown to significantly reduce the corticosterone levels in rats subjected to restraint stress [58] or a high-casein diet [54]. At the same time, oleuropein was also found to significantly increase the urinary noradrenaline and testicular testosterone levels in rats fed a high-casein diet [54]. Oleuropein aglycone was also shown to dose-dependently increase the plasma luteinizing hormone (LH) level in male Sprague-Dawley rats [54]. Cotreatment of olive leaf extract and oleuropein was also found to significantly increase the serum follicle-stimulating hormone (FSH) concentration in healthy mice [56]. In addition, olive leaf extract and oleuropein were found to protect mice from chemotherapeutic agent (cyclophosphamide and cisplatin)-induced testicular toxicity through the normalization of serum testosterone, LH, and/or FSH levels [56,57]. EVOO was also shown to significantly increase the serum testosterone and LH levels in healthy, young adult men consuming a controlled diet [55]. On the contrary, olive oil did not change serum sex hormone concentrations, including testosterone, LH, and FSH in Wistar rats [59], which may have been due to its blending ingredients.

Thyroid hormone is essential for body metabolism regulation, growth, and development [60]. The thyroid gland produces inactive thyroxine (T4) and biologically active triiodothyronine (T3). Thyroid follicular cells mainly produce and secrete T4 into the blood circulation as a prohormone [61,62]. Around 15–20% of T3 is directly synthesized and released from the thyroid gland [62]. The conversion of T4 to T3 requires deiodination catalysed by selenium-dependent iodothyronine deiodinase-1 (Dio1) and -2 (Dio2) [63,64]. Dio1 is present in several organs, like the liver, kidneys, and thyroid itself [63,64]. Dio2 is present in the adipose tissue, pituitary gland, and hypothalamus [63,64]. Dio2 is crucial for thyroid homeostasis as it converts T4 into T3 in the hypothalamus and pituitary gland, which serves as a negative feedback signal for the hypothalamic–pituitary–thyroid axis [65]. When the hypothalamus detects a low T3 level, thyrotropin-releasing hormone (TRH) is produced to stimulate subsequent secretion of thyrotropin or thyroid-stimulating hormone (TSH) in the anterior pituitary gland [61,62]. In addition to stimulation by TRH, TSH production is negatively regulated by cortisol [61]. Ultimately, TSH stimulates the thyroid follicular cells to increase thyroid hormone synthesis and levels [61].

To the best of our knowledge, a review that systematically summarizes the effects of olive on thyroid function is not yet available. Therefore, this review aims to summarise the effects of olive derivatives, including olive oil, polyphenols, and leaf extract, on thyroid function.

## 2. Materials and Methods

### 2.1. Literature Search Strategies

This systematic review was performed according to the Preferred Reporting Items for Systematic Reviews and Meta-Analyses (PRISMA) guidelines and checklist [66]. We conducted an electronic literature search using three databases—PubMed, Scopus and Web of Science—in October 2020. The search was performed using the following search string: (1) (olive OR tyrosol OR p-hydroxyphenylethanol OR p-HPEA OR hydroxytyrosol OR 3,4-dihydroxyphenylethanol OR 3,4-DHPEA OR oleocanthal OR p-HPEA-EDA OR 3,4-DHPEA-EDA OR oleuropein aglycon OR 3,4-DHPEA-EA OR ligstroside aglycon OR oleuropein) AND (2) (thyroid OR triiodothyronine OR T3 OR thyroxine OR T4 OR thyroid-stimulating hormone OR thyrotropin OR TSH OR thyrotropin-releasing hormone OR TRH). The search strategy is provided in Appendix A. The PRISMA checklist is included as Supplementary File S1.

### 2.2. Eligibility Criteria and Study Selection

We included cell culture, animal, and human studies reporting the effects of olive derivatives, including olive oil, extracts, or bioactive polyphenols, on thyroid hormones. The bibliographies of relevant research articles or review articles were read to identify potential articles missed during the database search. We excluded studies that were (1) only available in abstract form; (2) not written in English; and (3) books, book chapters, reviews, meta-analysis, conference/proceeding papers, letters to the editor, commentaries, and theses. The steps of the selection process, from identification to screening, eligibility, and inclusion of articles, are shown in Figure 1.

### 2.3. Study Extraction

Two reviewers (K.-L.P. and J.N.L.) independently extracted and screened the search results, first by referring to the titles and abstracts, followed by full-text screening. All articles that did not match the selection criteria were excluded. Discussions with the third reviewer (K.-Y.C.) occurred if there was a disagreement in article selection. The data extracted included the authors’ names, years of publication, types of olive treatment, study models, and findings. The information is tabulated as Table 1.

## 3. Results

### 3.1. Selection of Articles

From the literature search, 721 articles were identified, of which 188 were obtained from PubMed, 262 from Scopus, 268 from the Web of Science, and 3 additional articles from reference list checking. A total of 445 unique records were identified and screened after removing 276 duplicates. Four hundred and thirty-five articles were excluded based on the selection criteria, of which 13 articles were not written in English, 11 articles were not original articles, and 411 articles were not relevant to this topic. A total of 10 articles fulfilled all criteria mentioned, but the full text of one of the articles (published in 1977) was not available after multiple requests to the authors and publisher [76]. Therefore, only nine articles were included in this review.

### 3.2. Study Characteristics

The included studies were published between 2002 and 2019. All studies were conducted using animal models: rats [67,69,74,75], lambs [71], calves [72], goats [70], and chicken [68,73]. No in vitro or human studies on this topic have been conducted. Four studies employed olive leaf extract [67,68,73,74], while another three studies used olive oil [69,70,75]. Abd-Alla et al. [71] and Abdalla et al. [72] used solid olive pulp and solid pomace, respectively, which are residues from olive oil extraction or olive fruit squeezing. All included studies administrated the olive derivatives orally, i.e., through the diet, by oral gavage [68,69,70,71,72,75], or in drinking water [73,74]. Al-Qarawi et al. [67] did not disclose the details of oral administration, but it is assumed that oral gavage was used. The dosing and treatment conditions were heterogeneous. EVOO was used at a concentration of 0.5 g/kg body weight (approximately 137.5 mg for each 275 g rat) for 21 days [69] or 0.6 g/kg body weight daily (approximately 90–120 mg for each 150–200 g rat) for 28 days [75]. Farooq et al. [70] supplemented goats with 15 or 30 mL of commercially available olive oil (unknown specification) for 8 weeks. Olive leaf extracts in the lyophilized form were given to rats at a dose of 100–500 µg/day for 2 weeks [67]. A 500 mg/kg oleuropein-rich olive leaf extract (around 100 mg for one 200 g rat) or 150 mg/kg hydroxytyrosol-rich olive leaf extract (around 30 mg for one 200 g rat) was given in drinking water for an unknown treatment period and interval in the study by Mahmaudi et al. [74]. Oke et al. [73] used 5–15 mL/L of olive leaf extract (with at least 22–66 mg/L oleuropein) given ad libitum in drinking water for 8 weeks. Ahmed et al. [68] used a basal diet with olive leaf extract to achieve a final dose of 50–150 mg/kg oleuropein, and this was given to chickens for 18 weeks. On the other hand, 300 g of solid olive pulp was added to a basal diet and given to lambs for 3 months [71], and 15% solid olive pomace was added to a calf diet for 2 months [72]. The thyroid profile, including T3, T4, and/or TSH, was determined in healthy euthyroid animals [67,68,69,70] or animals under heat stress- [71,72,73] or with chemically-induced hypothyroidism [74,75]. TRH was not determined in any of the included studies.

### 3.3. Increased Thyroid Hormones in Euthyroid Animals

Olive oil and leaf extract exhibited thyroid-stimulating activities in healthy euthyroid animal models, including rats and livestock animals [67,68,69,70]. Supplementation with aqueous extract of olive leaves at 100–500 µg/day (oral intake for 14 days) dose-dependently increased the serum T3 level and reduced TSH in healthy male mature Wistar rats [67]. A similar increasing trend was observed for the T4 level, but it was not statistically significant [67]. In parallel, a basal diet with olive leaf extract (with a final concentration of 50–150 mg/kg oleuropein) also dose-dependently increased the plasma T3 levels in healthy Bandarah chicken [68]. Moreover, daily consumption of 15 or 30 mL of olive oil (commercially available but unknown specification) for 8 weeks significantly increased the serum T3 and T4 levels in adult male teddy goats [70]. However, the authors did not disclose essential information such as the specifications of the olive oil and supplementation route used [70].

On the other hand, supplementation with EVOO at a concentration of 0.5 g/kg decreased the plasma free T4 level but increased the free T3 level marginally in lactating Wistar rat dams [69]. EVOO supplementation did not alter the liver thyroid hormone receptor β1 (TRβ1) protein level or liver Dio1 mRNA expression in lactating dams [69]. The underlying mechanism of T4 reduction remains unknown, as the plasma TSH level was unchanged. The authors speculated that the low T4 level caused by EVOO supplementation might be due to T4 to T3 conversion by deiodinase from the peripheral tissues or organs like the kidneys (but not liver) [69]. Moreover, early EVOO supplementation in lactating dams increased the plasma free T3 level in mature male offspring due to neonatal programming during the 21-day breastfeeding period [69]. Similar to the lactating dams, the plasma free T4, TSH, liver TRβ1 protein level, and liver Dio1 mRNA expression levels were not significantly changed in mature offspring [69]. The higher T3 level in mature offspring could not be directly contributed to thyroid hormones in breast milk due to the lack of change in the total T3 level in breast milk during the 21-day breastfeeding period [69]. Additionally, the breast milk from EVOO-supplemented lactating dams did not alter the plasma free T3, T4, TSH, and liver TRβ1 protein levels or liver Dio1 mRNA expression in the breastfed offspring compared to control animals [69]. It is suggested that the bioactive compounds of EVOO, including olive polyphenols, are passed on to breastfed offspring via breast milk and subsequently reprogram their thyroid profiles upon maturation.

### 3.4. Improve Thyroid Profiles in Animals with Hypothyroidism

On the other hand, solid olive pomace residue, olive leaf extract, and olive oil have been demonstrated to have thyroid-stimulating and/or protecting activities in rats and livestock animals exposed to heat stress or chemically-induced hypothyroidism [71,72,73,74,75]. Abd-Alla et al. [71] reported that the consumption of a basal diet with 300 g of olive pulp residue for 3 months reduced chronic heat stress-mediated plasma T3 and T4 reduction among growing lambs during the summer season compared to the consumption of a rice straw diet. A similar finding was observed in a study in which heat-stressed female crossbred calves were supplemented with solid olive pomace (15% in the diet) for 2 months [72]. The olive pomace diet significantly restored the heat stress-downregulated plasma T3 level with simultaneous blood oxidative status improvements [72]. The serum malondialdehyde (MDA, a lipid peroxidation marker) level was significantly lower with a concurrently higher serum total antioxidant capacity and catalase (CAT) level in calves fed an olive pomace diet [72]. However, no further investigation was conducted to determine the nature of the relationship between the antioxidant and thyroid-stimulating activities of solid olive pomace. Oke et al. [73] also reported that drinking water with 15 mL/L of olive leaf extract ad libitum restored chronic heat stress-downregulated plasma T3 levels in Arbor Acre broiler chickens. Olive leaf extract also significantly improved the plasma oxidative status, as seen by a higher superoxide dismutase (SOD) level but lower MDA level [73]. Additionally, olive leaf extract was reported to contain a minimum of 4.4 mg/mL oleuropein [73], which was reported to have potent antioxidant activities [44].

Mahmoudi et al. [74] demonstrated the thyroid-stimulating and protecting properties of oleuropein- or hydroxytyrosol-rich olive leaf extracts following bisphenol A (BPA)-induced hypothyroidism. Drinking water containing 500 mg/kg oleuropein- or hydroxytyrosol-rich olive leaf extracts prevented BPA-mediated hypothyroidism in breastfed pups of BPA-treated lactating dams, as demonstrated by increased plasma free T3 and T4 levels and reduced TSH [74]. These olive leaf extracts also reduced the subsequent effects of BPA-induced hypothyroidism, especially on the body and bone growth suppression, as well as preventing pathological thyroid gland changes, including gland mass loss, follicular cell hypertrophy and dysfunction, and the reduction of the calcitonin-positive cell population [74]. These olive leaf extracts also improved the BPA-mediated total antioxidant capacity reduction in mature breast milk [74]. Moreover, oleuropein and hydroxytyrosol were detected in the breast milk of lactating dams [74].

In addition to olive leaf extracts, oral administration of EVOO (0.6 g/kg) was also shown to prevent deltamethrin (a pyrethroid insecticide)-mediated downregulation of TSH and T4 levels in adult female Wistar albino rats [75]. EVOO completely abrogated deltamethrin-induced body weight and absolute and relative thyroid weight reductions [75]. The oxidative status of thyroid tissues was significantly improved, as shown by lower levels of oxidation markers, including MDA and protein carbonyls, as well as higher levels of antioxidant proteins and enzymes, such as CAT, SOD, glutathione (GSH), glutathione peroxidase, and glutathione S-transferase [75]. However, the molecular mechanisms associated with the thyroid-stimulating and protecting activities of EVOO remain unclear.

## 4. Discussion

The current systematic review demonstrates the thyroid-stimulating properties of olive derivatives, including olive oil, olive leaf extract, and olive residues like pomace and solid pulp, which increase the concentrations of thyroid hormones (mainly T3) in euthyroid animals [67,68,69,70] and animals under heat stress- [71,72,73] or with chemically-induced hypothyroidism [74,75]. Several studies have consistently reported mutual improvements in thyroid function and systemic or thyroid tissue oxidative status upon supplementation with olive derivatives, especially in animals with hypothyroidism [72,73,74,75]. Chronic heat stress has been associated with the suppression of thyroid production due to thermoregulation to reduce endogenous heat production [77]. Additionally, chronic stress causes the upregulation of stress hormones like glucocorticoids, which subsequently inhibit the secretion of TSH [61,78]. As discussed earlier, olive leaf extract, olive pulp, and pomace residue were shown to effectively restore the concentrations of thyroid hormones during heat stress [71,72,73], but the mechanism by which this occurs remains unknown. Supranutritional supplementation with antioxidants like selenium and vitamin E has been found to effectively reduce oxidative stress and the respiration rate and body temperature of heat-stressed sheep [79]. Olive leaf extract containing a minimum level of 4.4 mg/mL oleuropein was also found to improve the plasma oxidative status [73], but the correlation between antioxidant activity and thermoregulation is unknown. In addition, consumption of a diet containing olive pulp was found to alleviate heat stress-induced reductions of plasma cortisol and cholesterol and the total lipid level in lambs [71]. T4 was positively and significantly associated with total cholesterol, low-density lipoprotein cholesterol, and high-density lipoprotein cholesterol levels in euthyroid men in a previous study [80]. Therefore, the higher cholesterol and total lipid levels in lambs supplemented with olive pulp could be attributed to the higher T4 level. Supplementation with olive pulp and pomace residue could be beneficial for the management of livestock with heat stress. Nevertheless, these products have very limited application in humans, because they are inedible for humans.

Hormone disruptors like BPA [81,82] and deltamethrin [83,84] are known to interfere with thyroid hormone production. BPA and other bisphenol analogues are structurally similar to the phenoxybenzene ring of the T3 hormone [85]. On the other hand, pyrethroids, including deltamethrin, are also structurally similar to thyroid hormones with the same phenoxybenzene ring as the side chain [83]. Both BPA [86,87] and deltamethrin [88] serve as thyroid receptor antagonists that competitively interfere with the binding between thyroid hormones and receptors. In addition to binding to thyroid hormone receptors, BPA also induces oxidative stress and subsequently affects thyroid hormone biosynthesis in thyrocytes by increasing hydrogen peroxide production, which can be abrogated by N-acetylcysteine (NAC, a precursor to GSH) [82]. Deltamethrin also induces oxidative stress in mice [89] due to its phenoxybenzene side chain, which has oxidant activity [90]. Antioxidants like lycopene, selenium, coenzyme Q10, GSH, vitamin C, and vitamin E have been reported to be effective against BPA or deltamethrin-induced oxidative stress and cellular damage and toxicity [91,92,93,94]. Olive oil, especially EVOO, contains high levels of polyphenols like oleuropein and hydroxytyrosol, which contributes to its potent antioxidant activity [8,44,95]. Olive leaves also contain similar polyphenols and phytochemicals to olive oil but with a much higher oleuropein concentration [41,42]. Olive leaf extracts could reduce oxidative stress in rats [96,97]. Daily consumption of 50 mL of oleuropein-rich EVOO could increase the plasma’s total antioxidant capacity and enzyme levels among the elderly [32]. As discussed earlier, supplementation with oleuropein- or hydroxytyrosol-rich olive leaf extract and EVOO was found to simultaneously improve thyroid function and improve the oxidative status of systemic or thyroid tissues following chemical-induced hypothyroidism [74,75]. In addition to restoring thyroid function, supplementation with EVOO also restored the FSH, progesterone, and oestradiol levels by reducing deltamethrin-induced ovary damages in adult female rats [75]. Moreover, olive leaf extract supplementation in lactating dams also alleviated BPA toxicity in rat pups via the bioactive olive polyphenols like oleuropein and hydroxytyrosol distributed in the breast milk [74]. Olive leaf extract also improved a BPA-mediated total antioxidant capacity reduction in breast milk [74]. However, there is no evidence to confirm the direct role of the antioxidant activities of EVOO or olive leaf extracts in protecting against chemically-induced hypothyroidism. Nevertheless, it is still rational to speculate that the antioxidant activities of EVOO and olive leaf extracts may protect the thyroid gland from BPA- or deltamethrin-mediated oxidative damage, subsequently contributing to thyroid protecting activities.

On the other hand, olive oil and leaf extracts were reported to have thyroid-stimulating properties. They were shown to increase the T3 level in euthyroid rats, chickens, and goats, with heterogeneous findings for the T4 level [67,68,69,70]. Olive oil was shown to significantly increase the serum T4 level in goats [70], while EVOO decreased the T4 level in lactating rats [69]. However, olive leaf extract did not significantly alter the T4 level in rats [67]. The T4 level of chicken was not determined after supplementation with oleuropein-rich olive leaf extract [68]. These contradictory observations have arisen due to the different olive derivatives and treatment conditions used. The thyroid-stimulating mechanisms of olive derivatives are largely unknown. Given the higher T3 level, olive derivatives might promote greater conversion of inactive T4 to biologically active T3 [67,69]. The current findings suggest that EVOO supplementation does not upregulate liver Dio1 mRNA expression [69], but its effects on other deiodinase enzymes located in the peripheral tissues or kidneys are still unknown. Although Dio1 is the major deiodinase involved in T4 to T3 conversion, the role of Dio1 in the euthyroid state is limited, as the general health and thyroid profiles of Dio1-knockout mice were not shown to be affected [98]. Dio2 is the primary enzyme that acts as the major source of plasma T3 in euthyroid rats and humans [99,100]. However, the role of Dio2 has not been determined in these studies. Additionally, all deiodinases are examples of selenoprotein, wherein they require selenium for their catalytic activity [60]. Selenium is a trace element with essential physiological functions, and defective selenoprotein biosynthesis or nutritional selenium deficiency can lead to reduce deiodinase activity and cause abnormalities in thyroid function [60,101,102]. In addition, supplementation with antioxidants like vitamin C [103,104] and soy isoflavones [105,106] but not α-tocopherol [107] could increase the selenium level, thereby affecting the activity of deiodinases. However, no studies have reported the antioxidant effects of olive derivatives on selenium homeostasis. In this review, the selenium status of animals consuming olive oil or olive leaf extract was not covered. Further study is required to elucidate the mode of action of olive oil and olive leaf extract related to the stimulation of the thyroid hormones, especially the involvement of Dio2 and selenium.

Several limitations should be noted for this systematic review. The current literature search was limited to the three largest multidisciplinary databases (PubMed, Scopus, and Web of Science). Unpublished and grey literature were not included. We also did not include all olive phytochemicals as search terms. Therefore, it is possible that less common olive phytochemicals were missed. We tried to overcome this limitation by checking the reference lists of included articles to identify additional studies. Additionally, a meta-analysis was not conducted due to the vast heterogeneity in olive derivatives, experimental models, and treatment conditions used in the respective studies. Lastly, the molecular mechanism by which olive derivatives stimulate thyroid hormones was not comprehensively discussed due to a lack of essential findings in the included articles.

## 5. Conclusions

Currently available in vivo studies have demonstrated the thyroid-stimulating properties of olive oil, olive leaf extract, and solid olive residue in euthyroid animals as well as in animals exposed to heat stress or with chemical-induced hypothyroidism. At present, human studies are needed to verify the effectiveness of these olive derivatives in improving the thyroid profile. The use of olive pulp and pomace residues could be beneficial in agriculture, like for livestock heat stress management, but they have minimal clinical applications in human health. Additionally, the molecular mechanisms of these olive derivatives involved in stimulating the thyroid hormones are largely unknown. There is a concurrent improvement of thyroid function and oxidative status in animals with hypothyroidism upon supplementation with olive derivatives; however, the causal relationship has not been determined. Olive oil and leaf extract are also postulated to induce a higher rate of conversion of inactive T4 to biologically active T3 in euthyroid animals. Further studies are required to elucidate the underlying molecular mechanism and the effectiveness of olive derivatives, especially olive oil and leaf extracts, via randomized controlled trials.

## Figures and Tables

**Figure 1 nutrients-13-00529-f001:**
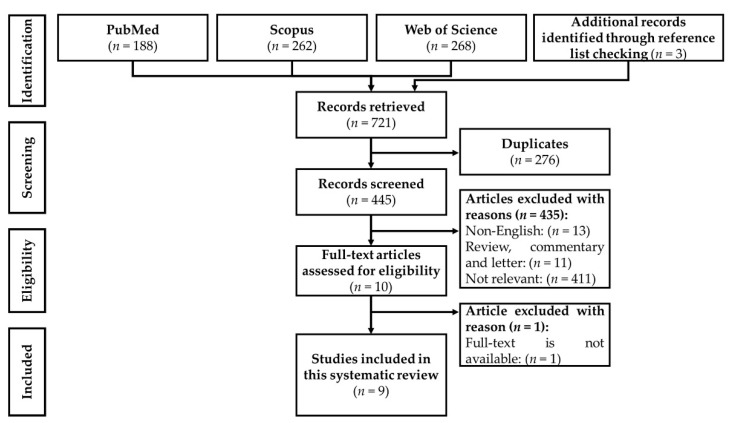
PRISMA flow chart of the systematic literature search.

**Table 1 nutrients-13-00529-t001:** Table of included studies.

References	Olive Derivative	Animal Model	Treatment Conditions	Findings
**Euthyroid animal studies**				
Al-Qarawi et al., 2002 [67]	Aqueous extract of *Olea europaea* leaves	Mature male Wistar rats (125–150 g; unknown sample size)	Control: distilled water (control) Treatment: 100, 250, and 500 µg/day of lyophilized olive leaf aqueous extract; oral gavage; 14 days	↑ serum T3 and ↓ TSH level dose-dependentlyNS for the serum T4 level.
Ahmed et al., 2017 [68]	Oleuropein-rich olive leaf extract	Bandarah chickens (24 weeks old, *N* = 144)	Control: basal diet (control) Treatment: olive leaf extract in the basal diet, standardized to 50, 100, and 150 mg/kg of oleuropein; 18 weeks	All doses significantly and dose-dependently ↑ plasma T3, ↑ plasma SOD and total antioxidant capacity with ↓ MDA
Quitete et al., 2018 [69]	EVOO from Santa Cruz Biotechnology, Inc., TX, USA; Cat. No.: sc-215631A; Lot No.: #D1814)	Wistar rat dams (*N* = 40) (measured up to 21 days during lactation) and the male pups (*N* = 240) (measured up to 180 days postnatal)	Control: soybean oil (control) Treatment: EVOO, fish oil or coconut oil at 0.5 g/kg body weight during the 21-day lactation period.	EVOO ↓ plasma free T4 levels in lactating dams significantly more than in the control (NS for plasma free T3, TSH, liver TRβ1 protein levels and liver Dio1 mRNA expression). No alteration of the breast milk total T3 levels in EVOO group vs. control. NS for plasma free T3, T4, TSH, and liver TRβ1 levels, and liver Dio1 mRNA expression in 21-day breastfed pups in EVOO group vs. control. ↑ plasma free T3 level of mature offspring in the EVOO group vs. control (NS for plasma free T4, TSH, liver TRβ1 protein level, and liver Dio1 mRNA expression).
Farooq et al., 2019 [70]	Commercially available olive oil (unknown specification)	Adult male teddy goats (*N* = 9)	Control: basal diet Treatment: 15 and 30 mL olive oil (probably in diet); 8 weeks	↑ serum T3 (both 15 and 30 mL) vs. control. ↑ serum T4 (only 30 mL) vs. control.
**Hypothyroid animal studies**				
Abd-Alla et al., 2007 [71]	Solid olive pulp from North Sinai, Egypt	Chronic heat-stressed growing lambs (*N* = 15)	Control: basal diet with rice straw/green acacia leaves Treatment: basal diet with 300 g of olive pulp solid; 3 months	Restored chronic heat-downregulated plasma T3 and T4 levels vs. control.
Abdalla et al., 2015 [72]	Solid olive pomace from new EL-Salheya olive mill factory—Sharkia Governorate, Egypt	Heat-stressed female crossbred (Brown Swiss x Baladi) calves (*N* = 10; 8-10 months old; mean body weight 112 kg)	Control: basal diet Treatment: olive pomace (15% of the diet); 2 months	↑ serum T3 levels, ↓ MDA level, ↑ total antioxidant capacity and ↑ CAT levels significantly vs. control.
Oke et al., 2017 [73]	Olive leaf extract manufactured by Olive Leaf Australia (Coominya, Australia) with at least 4.4 mg/mL oleuropein	Chronic heat-stressed Arbor Acre broiler chickens (*N* = 240)	Control: drinking water without extract. Treatment: drinking water with 5, 10, and 15 mL/L leaf extract; 8 weeks.	↑ plasma T3 level with ↑ plasma SOD level (only 15 mL/L extract). ↓ plasma MDA level (only 15 mL/L extract). Leaf extract (5–15 mL/L) is non-hematotoxic.
Mahmoudi et al., 2018 [74]	Oleuropein- and hydroxytyrosol-rich *Olea europaea* leaves extracts	Lactating adult female Swiss strain rats (*N* = 16; ~200 g) with female (*N* = 64) and male pups (*N* = 64). Outcomes were observed in the breastfed pups until 20 days of age.	BPA control: 250 mg/kg BPA; intramuscular injection of dams Treatment: BPA + oleuropein-rich extract (500 mg/kg body weight) or hydroxytyrosol-rich extract (150 mg/kg body weight) in drinking water for dams	Both extracts restored the total antioxidant capacity of BPA-reduced mature breast milk from lactating dams. Both extracts prevented BPA-induced upregulation of the TSH level and the reduction of plasma free T3 and T4 levels in breastfed pups. Both extracts prevented BPA-mediated thyroid gland mass loss and pathological changes (follicular cell hypertrophy, follicular cell dysfunction and calcitonin-positive cell number and area reduction) in pups. Both extracts improved BPA-suppressed body growth and bone health in pups.
Mekircha et al., 2018 [75]	Extra virgin *Oleo europaea* L. oil by traditional first cold pressure extraction on healthy *Olea europaea* L. fruits	Adult female Wistar albino rats (*N* = 30; 150–200 g)	Deltamethrin control: deltamethrin (2.56 mg/kg; oral) Treatment: EVOO (0.6 g/kg, oral) with or without deltamethrin for 28 days	Abrogated deltamethrin-induced body weight loss and the reduction of the absolute and relative thyroid weight. Restored deltamethrin-downregulated TSH and T4 levels to the baseline. ↓ deltamethrin-mediated oxidative stress in thyroid tissues with ↓ MDA, ↓ protein carbonyls, ↑ GSH, ↑ CAT, ↑ glutathione peroxidase, ↑ SOD and ↑ glutathione S-transferase levels.

Abbreviations: ↑, increase or upregulate; ↓, decrease or downregulate; BPA, bisphenol A; CAT, catalase; Dio1, iodothyronine deiodinase 1; EVOO, extra virgin olive oil; GSH, glutathione; MDA, malondialdehyde; *N*, sample size; NS, not significant; T3, triiodothyronine; T4, thyroxine; TRβ1, thyroid hormone receptor β1; TSH, thyroid-stimulating hormone; SOD, superoxide anion dismutase; vs., versus.

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
