# Peer review of "Thyroid-Modulating Activities of Olive and Its Polyphenols: A Systematic Review"

_nutrients, 2021, doi:10.3390/nu13020529_

Round 1

Reviewer 1 Report

The manuscript is a review that systematically summarizes the effects of olive derivatives, including olive oil, polyphenols and leaf extract, on thyroid function. The paper describes a very interesting, credible approach to indicate only those publications in which the aim was to show the influence of olive on the functioning of the thyroid gland. The results of the literature analyzes (9 out of 445) showed the lack of such studies on the human population. The results presented in the manuscript are extremely helpful in outlining the purpose of future research and to work out methodology.

In general the text is well written, but especially Results chapter needs improvement (dateils below).

Comments:

line 21-22:

In the abstract isn’t clear what this phrase “… 9 articles that met the inclusion and exclusion criteria” means. 

Line 150:

Table 1 requires improvement – data presented in the table are also described in the text (lines 166-263.

In my opinion table 1 should be more readable, data should be presented in clear way, articles could be sorted i.e. by animal model, treatment condions anf findings should be presented in order.  

Line 191:

TRH? Please explain this abbreviation.

Author Response

Dear reviewer,

Thank you for reviewing our manuscript. We appreciate the constructive comments provided and have responded to each of them in the attached response sheet. Changes in the text are highlighted in yellow. 

We hope that the revised manuscript can met the standard of the esteemed journal. We look forward to receiving your positive reply.

Reviewer 2 Report

This is an interesting review of the literature relating olive oil and leaf extract to measures of thyroid function in animal models. 

The way data was collected is well described.

The paper is well organized, but may be improved with attention to the Table 1. I would suggest modyfying the style of Evidence table. The paragraphs desciribing included studies should be clearly separated. Now the text in ‘Animal model and treatment condition’ and ‘Findings’ in reference 70 and 71 blends together, making it difficult to read - at first glance.

I understand that without knowing the mechanisms underlying those observations, you can only make assumptions, but I'm not sure if too much space has been devoted to selenium  (lines 340 -371) while the authors themselves indicated that In this review, the selenium status of animals consuming olive oil or olive leaf extract is not determined’

Iron, vitamin A are also very important in thyroid metabolism, but the title of the article is  Thyroid-modulating activities of olive and its polyphenols: A systematic review.

Author Response

Dear reviewer,

Thank you for reviewing our manuscript. We appreciate the constructive comments provided and have responded to each of them in the attached response sheet. Changes in the text are highlighted in yellow.

We hope that the revised manuscript can meet the standard of the esteemed journal. We look forward to receiving your positive reply.

Reviewer 3 Report

The authors did a comprehensive review of the literature about thyroid function and the effects of different olive oil derivatives.

The review article is well written, and I will only suggest correcting some minor errors:

*line 82: "than in olive oil"

*line 157: 721 instead of 720

*line 175: details instead of detailed

*line 381: supplementation instead of supplemented

*Table 1: Did you follow any order for the references? Maybe it could be helpful to introduce them in chronological order.

Author Response

(The authors gave the same response as above.)
